# Resveratrol Protects Cardiac Tissue in Experimental Malignant Hypertension Due to Antioxidant, Anti-Inflammatory, and Anti-Apoptotic Properties

**DOI:** 10.3390/ijms22095006

**Published:** 2021-05-08

**Authors:** Jelica Grujić-Milanović, Vesna Jaćević, Zoran Miloradović, Djurdjica Jovović, Ivica Milosavljević, Sladjan D. Milanović, Nevena Mihailović-Stanojević

**Affiliations:** 1Laboratory for Experimental Hypertension, Institute for Medical Research, Department for Cardiovascular Research, University of Belgrade, National Institute of the Republic Serbia, 11000 Belgrade, Serbia; zokim@imi.bg.ac.rs (Z.M.); djurdjica@imi.bg.ac.rs (D.J.); nevena@imi.bg.ac.rs (N.M.-S.); 2Department for Experimental Toxicology and Pharmacology, National Poison Control Centre, Military Medical Academy, 11000 Belgrade, Serbia; v_jacevic@yahoo.com; 3Medical Faculty of the Military Medical Academy, University of Defence, 11000 Belgrade, Serbia; 4Department of Chemistry, Faculty of Science, University of Hradec Kralove, 500 30 Hradec Kralove, Czech Republic; 5Institute of Pathology and Forensic Medicine, Military Medical Academy, 11000 Belgrade, Serbia; imilosavljev@gmail.com; 6Institute for Medical Research, Department for Biomedical Engineering and Biophysics, University of Belgrade, National Institute of the Republic Serbia, 11000 Belgrade, Serbia; sladjan.milanovic@imi.bg.ac.rs

**Keywords:** resveratrol, spontaneously and malignant hypertension, heart

## Abstract

Hypertension is one of the most prevalent and powerful contributors of cardiovascular diseases. Malignant hypertension is a relatively rare but extremely severe form of hypertension accompanied with heart, brain, and renal impairment. Resveratrol, a recently described grape-derived, polyphenolic antioxidant molecule, has been proposed as an effective agent in the prevention of cardiovascular diseases. This study was designed to examine chronic resveratrol administration on blood pressure, oxidative stress, and inflammation, with special emphasis on cardiac structure and function in two models of experimental hypertension. The experiments were performed in spontaneously (SHRs) and malignantly hypertensive rats (MHRs). The chronic administration of resveratrol significantly decreased blood pressure in both spontaneously and malignant hypertensive animals. The resveratrol treatment ameliorated morphological changes in the heart tissue. The immunohistochemistry of the heart tissue after resveratrol treatment showed that both TGF-β and Bax were not present in the myocytes of SHRs and were present mainly in the myocytes of MHRs. Resveratrol suppressed lipid peroxidation and significantly improved oxidative status and release of NO. These results suggest that resveratrol prevents hypertrophic and apoptotic consequences induced by high blood pressure with more pronounced effects in malignant hypertension.

## 1. Introduction

Cardiovascular diseases represent the first cause of mortality in the Western world and, due to this, particular attention is paid to them in both the scientific world and in the public opinion. Epidemiological studies have demonstrated that hypertension, as a significant cardiovascular risk factor, represents the main cause of death and, in the general population, it affects from 30% to 45% of people, with a tendency to increase incidence from the age of 50. Malignant hypertension is a particularly severe form of arterial hypertension defined by extremely high systolic and diastolic arterial pressure (SAP > 200 ± DAP > 130 mmHg) [1]. In the general population, the prevalence of malignant hypertension is relatively low, but it had an extremely severe prognosis considering ≈80% mortality in 2 years [2]. Despite the serious character of malignant hypertension, it is not studied enough.

The spontaneously hypertensive rats (SHRs) are widely used as a rat model of primary or essential hypertension. They have a similar vascular anatomy and structure as humans characterized by altered vascular tone, increased vascular contractility, and long-term, stable cardiovascular structural remodeling [3,4]. Malignant hypertension can develop from essential hypertension through serious long-term inhibition of endothelium-derived, vasodilatory, nitric oxide (NO). In particular, persistent N^G^-L-Arginine Methyl Ester (L-NAME) (Figure 1B) inhibition of NO biosynthesis in SHRs leads to a significant increase in blood pressure followed by profound vasoconstriction, oxidative stress, and structural alterations of the conduit and large arteries, as well as cardiac hypertrophy [4,5,6].

Untreated high blood pressure can contribute to vascular remodeling via structural and functional changes within the arterial wall [7,8] and induced endothelial dysfunction has been attributed to reducing NO bioavailability [8]. Reduced NO availability and vasoconstriction in the hypertensive condition induces oxidative misbalance. Oxidative stress is defined as “an imbalance between oxidants and antioxidants in favor of the oxidants, leading to a disruption of redox signaling and uncontrolled reactive oxygen species (ROS) production” [8]. ROS induce protein oxidation and dysregulate cell signaling, leading to inflammation, proliferation, fibrosis, and apoptosis, which are important processes contributing to impaired vascular function and cardiovascular remodeling in hypertension [9]. Additionally, hypertension has been shown to increase neutrophil activation [10]. Neutrophils and monocytes normally produce and release myeloperoxidase (MPO) [11]. Elevated level of MPO reflects the degree of neutrophil activation and it has been associated with vascular dysfunction [12]. Additionally, MPO is involved in the formation of ROS and the oxidation of biological material [13]. The above findings suggest that elevated levels of MPO could be observed in hypertension.

Transforming growth factor-beta (TGF-β) is a locally generated cytokine that plays a mediator role in promoting alterations in vessel structure [14]. It is found in all cell types of the cardiovascular system, including cardiac myocytes, vascular smooth muscle, endothelial cells, fibroblasts, and blood cells [15]. The endothelial cell dysfunction in hypertension relates to the abundance of TGF-β in the arterial wall, which promotes intimal proliferation as well as the development of vascular lesions accompanied by apoptosis in the vascular smooth muscle cells [5].

The Bcl-2 proteins can localize to different organelles including the endoplasmic reticulum, the Golgi apparatus, the nuclear outer membrane, or the nucleus itself [16]. It plays a role in the inhibition of apoptosis [17]. On the other side, Bax, a member of the Bcl-2 family, promotes apoptotic death. The ratio of Bcl-2 to Bax determines survival or death after an apoptotic stimulus [16]. Thus, we speculated that Bcl-2 or Bax overexpression can induce elevated blood pressure and may also cause myocyte death in hearts of rat in different models of hypertension.

More scientists highlighted that some components of the diet, including polyphenols, show a protective effect on the onset of cardiovascular disease (CVD) [18]. Several studies have shown that low to moderate wine intake had been associated with lower mortality from cardiovascular and cerebrovascular diseases [19,20]. After these observations, a great deal of attention has been paid to red wine [21]. The cardioprotective effects of red wine have been attributed to resveratrol [22]. Resveratrol (*trans*-3,4′,5-trihydroxystilbene, C_14_H_12_O_3_) (Figure 1B) is a natural polyphenol found in many plant species including grapes and a variety of berries. Several studies reported that resveratrol possessed antioxidant [23], anti-inflammatory [24], and proapoptotic activity [25]. Dietary polyphenol, resveratrol, has potential to improve endothelial dysfunction and decrease overall blood pressure [5,26]. In the hypertensive condition, resveratrol decreases vascular oxidative stress by scavenging H_2_O_2_ and preventing oxidative stress-induced endothelial cell death [27]. It is found that it may activate endogenous defense systems and regulate cellular signaling processes [28].

Considering the aforementioned information, resveratrol seems to serve as an effective dietary supplement for the protection and preservation of the aorta in different hypertensive conditions. Therefore, this study has been designed to examine chronic resveratrol administration on cardiac structure and function and correlate it with oxidative stress and inflammation status of heart in the spontaneously (SHRs) and malignant hypertensive (MHRs) rats.

## 2. Results

### 2.1. Body and LV and RV Weights

After the treatment, period body weight (b.w.) was comparable between all experimental groups (Table 1). However, both resveratrol treated SHRs and MHRs had significantly lower body weights compared to vehicle treated control rats (*p* < 0.001; *p* < 0.05).

Left (LV) and right ventricular (RV) weight and ventricular weight indexes (LVI; RVI) were significantly higher in MHRs compared to SHRs (Table 1). After resveratrol treatment, there were significant differences in LV and RV weights and ventricular weight indexes compared to their respective controls (Table 1).

### 2.2. Hemodynamic Study

The data collected by hemodynamic measurement showed that systolic arterial pressure (SAP) was 176.3 ± 4.69 mmHg in SHRs (Figure 1). Additionally, after 4 weeks of blocked NO synthesis in MHR, SAP raised to 201.12 ± 1.74 mmHg. The application of resveratrol significantly reduced this increase in both SHR and MHR groups (*p* < 0.001). The cardiac output (CO) was significantly decreased, and total vascular resistance (TVR) was significantly increased in the MHR compared with control groups (*p* < 0.001). Compared to their respective controls, the CO increased significantly in the SHR + R and MHR + R groups (47% to 29.4%, respectively) after resveratrol treatment. However, chronic resveratrol treatments reduced TVR in both treated resveratrol groups. The heart rate (HR) was not different between experimental groups. However, in malignant hypertensive rats, resveratrol increased CO and reduced SAP as well as TVR almost to the value of the control group.

### 2.3. Redox State

The thiobarbituric acid reactive substances (TBARSs) were significantly increased in the MHR group, which had significantly higher advanced oxidation protein products (AOPPs) compared to control group (Figure 2; *p* < 0.001; *p* < 0.01; respectively). Resveratrol treatments induced a significant reduction in TBARSs as well as AOPP level in both treated groups, SHR + R (*p* < 0.05, *p* < 0.01) and MHR + R, in comparison to their respective controls (*p* < 0.001, *p* < 0.001).

In heart tissue of MHRs, the level of thiol groups increased significantly, by 30%, compared with the SHR group (Figure 2). In the same MHR group, the superoxide anion radical (O_2_^−^) and prooxidative balans (PAB) were significantly higher (*p* < 0.001, *p* < 0.001) and the NO_2_ levels were significantly decreased (*p* < 0.01) compared with control. Interestingly, the application of resveratrol improves redox status in both hypertensive groups. Markedly enhanced the nitrite (NO_2_) levels (*p* < 0.001) and attenuated O_2_^−^ as well as thiol groups were seen in the SHR animals that received resveratrol (*p* < 0.001, *p* < 0.05). A similar trend was maintained in the MHR + R group, the NO_2_ was significantly higher (*p* < 0.001), but the level of superoxide anion and thiol groups were markedly lower (*p* < 0.001, *p* < 0.001) compared to MHR. As expected, these parameters were significantly better in SHR + R compared to MHR, and in MHR + R it was almost the value of the control group.

The blockade of NO syntheses decreased superoxide dismutase (SOD), catalase (CAT), and glutathione peroxidase (GPx) activities in MHR compared to SHR (Figure 3). After 4 weeks, application of resveratrol significantly prevented the reduction in antioxidant enzyme in SHR + R (*p* < 0.05; *p* < 0.01; *p* < 0.001) and MHR + R (*p* < 0.01; *p* < 0.001; *p* < 0.01). As expected, resveratrol revealed significant improvement activity of antioxidant enzyme as compared with MHR and its values showed no relevant differences compared with control.

At 4 weeks, L-NAME treated SHRs had a significantly elevated level of MPO compared with vehicle treated control rats (Figure 4, *p* < 0.01). As expected, resveratrol significantly reduced MPO (*p* < 0.001) compared to SHR, as is the case in MHR + R compared to MHR (Figure 4, *p* < 0.01). In the SHR + R group, the decrease in MPO was higher as compared with MHR (*p* < 0.001), and the MHR + R group was similar to the SHR group.

### 2.4. Histopatological Parameters

The high blood pressure in SHRs leads to the partial vacuolar degeneration of the endocardium and myocardium in hypertrophied myocytes, the altered muscle fibers partially lost their transverse striations, and the cytoplasm is filled with clearly visible vacuoles of various shapes, sizes, numbers, and arrangements, as well as the nuclei, which are extremely basophilic, large without visible nucleoli. Histological changes were more intensive in MHR, with markedly hypertrophied myocytes, in which the cytoplasm is filled with numerous, clearly visible vacuoles, the nucleus is hyperchromatic without nucleolus, and in muscle fibers, the transverse striation is mostly lost. The application of resveratrol significantly ameliorates the morphological changes caused by hypertension. However, in the SHR + R group, mild hypertrophy of the myocytes with hypochromatic cytoplasm, tiny, single, round vacuoles, and large oval nuclei was observed. Additionally, in MHR + R, resveratrol treatment significantly diminished structural changes in the heart (Figure 5).

In edematous myocytes, the cytoplasm is filled with large, clearly visible vacuoles, of various shapes, numbers, and arrangements. The nuclei of these cells are large, round, or oval and distinctly basophilic. Cardiomyocyte damage score significantly decreased after chronic treatment with resveratrol in both treated groups, SHR + R and MHR + R, in comparison to their respective controls (*p* < 0.001). As expected, this parameter was significantly better in SHR + R compared to MHR, and its values showed no relevant differences in the MHR + R group (Figure 4B).

Immunohistochemistry of the heart tissue SHR showed that TGF-β was present, with a granular appearance, in the cytoplasm. The overexpression of TGF-β was in the myocyte of MHR. Treatment with resveratrol induced a total absence of TGF-β, particularly in the SHR + R group, and it was partly present in single myocytes of MHR + R (Figure 6).

The protein expression of Bax in heart tissue was in the peripheral parts of the myocytes in the SHR group. In the MHR dark-brown immunoreactive Bax products were seen in the cytoplasm of whole cells and the nucleus is difficult to see or not notice. However, after resveratrol treatment in the SHR + R group, positive Bax immunoreactivity was immunohistochemically demonstrated only of the wall of myocytes, and this expression was diffusely distributed in the myocytes of the MHR + R rats (Figure 7).

In the SHR + R group, some myocytes showed positive Bcl-2 immunoreactivity. Slight positive Bcl-2 immunoreactivity was seen in myocytes of SHR and MHR + R. There were no findings of positive Bcl-2 immunoreactivities in the myocytes of MHR (Figure 8).

## 3. Discussion

Key observations from this study are as follows: (1) MHRs compared to SHRs have significant impairments of hemodynamic parameters, oxidative stress, and inflammatory status that are accompanied by more intensive histological changes in the heart tissue and (2) resveratrol administration on SHRs, as well as MHR animals, elicits cardioprotection, and improves hemodynamics and histopathology parameters, which can be attributed to antioxidant, anti-inflammatory, and antiapoptotic properties of the resveratrol.

Our results indicate that, in a genetic model of hypertension, changes in redox and inflammatory status compromise cardiovascular function and alter hemodynamic parameters in a similar way as shown in the study of Senoner and Dichtl [29]. The L-NAME-induced malignant hypertension dysregulates production of O_2_^−^ and, together with the interplay between O_2_^−^ and NO, contributes to an altered cellular redox status and oxidative damage of myocytes in heart tissues. It seems that heart damage is not only related to permanently and rapidly increased blood pressure itself (the hemodynamic stress), but also to the underlying adverse biochemical alteration (biochemical stress), and, on the other hand, to the ability of the heart to withstand with this stress.

In the present study, the results indicate that changes in oxidative status in the SHR are equally important in the development of hemodynamics and pathological alterations, clearly supporting results from many authors [9,29]. Recent studies have demonstrated that high blood pressure is a primary and important factor that causes the development of cardiac hypertrophy [30]. Additionally, some investigations have shown the central role of ROS in vascular function and inflammation in hypertension [29,31]. Therefore, malignant hypertension, in our study, was accompanied by a reduction in NO bioavailability, resulting from reduced NO production and/or increased NO degradation by superoxide anion, leads to an additional increase in blood pressure and pathological alternation of cardiac tissue. The investigation of Liu [32] considered NO molecules as a factor that mediates the vasodilator activity of resveratrol. Similar results were observed in different animal models of hypertension [3]. Previously, we have shown resveratrol to decrease the high blood pressure and to prevent remodeling of the aorta walls of SHRs and MHRs [5]. In this study, resveratrol in both models of hypertension reduced O_2_^−^ and increased the level of the vasodilator NO, which protects against high blood pressure and subsequent cardiac hypertrophy. We have shown that elevated blood pressure declined by 18.2% in SHRs and 16.7% in MHRs after chronic resveratrol treatment. Generally, chronic pressure overload is characterized by an increase in total vascular resistance, which was confirmed in both our hypertensive groups. This value was significantly reduced in both resveratrol treated groups, which could indicate that this polyphenol is a potential enhancer of vascular function. These results give hope that resveratrol could be a great ally in all degrees of hypertension, especially in association with the standard FDA recommended drugs. Actually, pharmacological intervention studies have shown that the reduction of just 10% of the values of the blood pressure resulted in a 40% reduction in mortality from cardiovascular accidents and 16%–20% from coronary accidents.

Although, it is well-known that lipide peroxidation has been increased in hypertensive rats [28]. The attenuation of antioxidant activity, as well as accumulation of protein oxidation and lipid peroxidation products in hypertensive models, indicate that there is an excessive amount of reactive oxygen species and a reduction in antioxidant activity, which is consistent with the results of other authors [33]. Several studies have shown that oxidative stress plays a pivotal role in mediating the production and secretion of cytokines [11,34], thus linking ROS with inflammation, endothelial activation, and dysfunction. The mechanisms by which oxidative stress elicits endothelial cell injury during hypertension are likely to include induction of inflammatory processes and activation of cellular apoptotic pathways [29]. In the present study, MHRs had both exacerbated levels of lipids and protein oxidative products, as well as significantly reduced antioxidative potential concerning the SHR. Resveratrol is known to promote antioxidant defenses by regulating a host of antioxidant enzyme [35,36]. Additionally, resveratrol improved oxidative damage in vasculature, heart, skeletal muscle, kidney, and brain tissues caused by high blood pressure [23,36,37,38,39].

In our study resveratrol activates the antioxidant system, including superoxide dismutase, glutathione peroxidase and catalase, leading to a decrease in the prooxidative balans. We have found that the activity of the antioxidant enzymes was markedly increased after treatment with resveratrol in both SHRs and MHRs. An increase in antioxidative enzyme production caused by resveratrol was involved to some degree in the mechanism by which resveratrol positively modulates blood pressure. Consequently, in this study, resveratrol could exert a positive effect on heart tissue via a direct modification of antioxidant defense or by decreasing blood pressure in both SHRs and MHRs.

Eiserich et al. [40] have shown that MPO diminishes the bioavailability of NO and impairs NO-dependent vessel relaxation, which is confirmed in our results for both SHR and MHR models. As described previously, NO has an important role in vascular tone [41]. In healthy subjects, NO causes vasodilation of the blood vessels. In contrast, hypertension decreases bioavailability of NO, attenuating vasodilatation of peripheral blood vessels and remarkably elevating vasoconstriction of conduit arteries [1]. In this study, we assume that resveratrol treatment in both SHRs and MHRs improved NO bioavailability, at least in part, through MPO inhibition. Therefore, our study, for the first time, shows effects of resveratrol on the MPO activity in heart tissue, particularly in the model of malignant hypertension, as well as the beneficial effects of resveratrol treatment in this model.

Elevated blood pressure induces structural changes in the vasculature, including hypertrophy and overproduction of TGF-β. In different models of the arterial hypertension in aorta tissue, an increased expression of TGF-β has been reported [5,42]. The previously published papers have shown that TGF-β and bone morphogenetic protein pathways may regulate the change in expression of ~19% of hypertension-responsive genes [43]. Otherwise, uncontrolled ROS production leads to increased expression and secretion of TGF-β [44]. In the presented study, myocytes of MHRs showed TGF-β overexpression, compared with the degree of expression in myocytes of control hearts. Overexpression of TGF-β in the heart tissue of the MHRs promotes hypertrophy of the myocytes and an increase in the number of microvessels. These processes result in rupture and degradation of myofilaments accompanied with cardiac apoptosis. Resveratrol had a maximal effect in reversing them in SHRs. Additionally, in malignant hypertensive animals, treatment with resveratrol led to a tissue structure that was almost identical to that of the untreated controls. The overexpression of TGF-β was rare in salvaged myocytes of the SHRs, but it was moderately frequent in heart tissue of malignant hypertensive rats after resveratrol treatment.

Hockenbery et al. [16] found that Bcl-2 protein is not expressed in normal cardiac tissues. Some research conformed Bcl-2 as a prototype for an antideath or survival factor [17]. In this study, there was no evidence of positive immunoreactivity in the hearts of MHRs, as described previously [45]. Positive Bcl-2 immunostaining in myocytes after resveratrol treatment promotes this polyphenol as an antiapoptic factor.

The Bax protein is found in various tissues [46]. Additionally, it is a member of the Bcl-2 family and, when overexpressed, accelerates the apoptotic death induced by cytokines [17]. This study showed Bax overexpression in the myocytes of MHRs, while it was rarely present in the SHRs. After resveratrol treatment, there is no Bax expression in the myocytes of SHRs, but it is expressed only in the bordering regions of myocytes of the MHRs. Our study revealed that in the salvaged myocytes after resveratrol treatment, Bcl-2 protein is expressed, and overexpression of Bax is rare. These data suggest that Bcl-2 activation could impede Bax, thus reducing cell apoptosis. Therefore, it is reasonable to speculate that resveratrol might be a potential therapeutic target to reverse vascular remodeling and retard the progression of hypertrophy changes in heart tissue of hypertensive animals.

Our results show that resveratrol possesses anti-inflammatory and antiapoptotic properties in both models of hypertension, with more pronounced effects in malignant hypertensive rats. Actually, resveratrol treatment normalized the myocardial ultrastructure, as evidenced by intact myofibrils and a reduction in number of vesicula in myocytes. This effect of resveratrol on cell survival may be attributed to multiple mechanisms, including prevention of cardiomyocyte apoptosis, regulation of inflammation, and reduction in oxidative stress.

## 4. Materials and Methods

### 4.1. Materials

All the environmental conditions, as well as procedures adopted for housing and handling of experimental animals, were in strict compliance with Guidelines for Laboratory Animal Welfare, Ethics Committee for Experiments on Animals of the Institute for Medical Research (IMR), University of Belgrade, Serbia, which was adopted by the current National Law on Animal Welfare (Serbian Gazzette, No. 41/2009). The study protocol was approved by the Ethics Committee for Experiments on Animals issued by IMR (No. 0316-1/11).

The experiments were performed on 6-month-old female SHRs (200 to 250 g) bred at the IMR. They were descendants of breeders originally obtained through Taconic Farms, Germantown, NY, USA. The experimental animals were housed in groups of four in plastic cages (Macrolon^®^ cage type 4, Bioscape, Castrop-Rauxel, Germany) with sawdust bedding (Versele-Laga, Deinye, Belgium) certificated as having contaminant levels below toxic concentrations. The environmental conditions were controlled and monitored by a central computer-assisted system with a temperature of 22 ± 2 °C, a relative humidity of 55% ± 15%, 15–20 air changes/h, and artificial lighting of approximately 220 V (12 hrs light/dark cycle). The experimental animals had free access to food, commercial pellets for rats (Veterinarski Zavod Subotica, Victoria group, Subotica, Serbia) and tap water from municipal mains, filtered through a 1.0 µm filter (Skala Green, Subotica, Serbia).

All animals were examined for control values of systolic blood pressure by an indirect method using a tail-cuff, pneumatic pulse detector and direct recorder (Physiograph Four, Narco Bio-System, Houston, TX, USA). After ensuring that all animals had been hypertensive, they were divided into four groups according to the 4-week drug treatment. The first group was SHR (*n* = 12), who received tap water 0.5 mL by gavage. The second group was SHR + R group (*n* = 12), who were treated with resveratrol (Sigma-Aldrich, St. Louis, MO, USA) with a dose of 10 mg/kg/day by gavage. The malignant hypertensive group (MHR) represents the SHRs that were treated with L-NAME (Sigma-Aldrich, St. Louis, MO, USA), a nonselective inhibitor of NO synthase enzyme dissolved in tap water in a dose of 10 mg/kg/day (*n* = 12). The fourth group was MHR + R, SHRs (*n* = 12) treated with the combination of L-NAME and resveratrol in the same dosages (Figure 1A).

### 4.2. Hemodynamic Measurements

At the end of treatment, all animals were anesthetized (35 mg/kg *ip* sodium pentobarbital) and then hemodynamic and biochemical studies were performed. SAP and HR were measured directly through a femoral artery catheter (PE–50, Clay-Adams Parsippany, NJ, USA) using a low volume displacement transducer P23 Db (Statham, Oxnard, CA, USA), and recorded on a direct writing recorder. Additionally, their left carotid arteries and right jugular veins were prepared for CO measurement using a previously described [47] modification of Coleman’s application of the dye dilution technique. Indocyanine green was used as the indicator, and a recording densitometer (Beckman Instruments, Columbia, MD, USA) was utilized for the determination of dye in the blood and registration of the dilution curve. TVR was calculated by dividing the mean arterial pressure by CO and expressed as mmHg × min × kg/mL.

### 4.3. Redox State of Heart

Hearts were harvested, weighed, and stored at −70 °C for later analysis. TBARSs, as a marker of lipid peroxidation, were measured by using 4,6-dihydroxy-2-mercaptopyrimidine according to the method of Ohkawa et al. [48]. AOPP were measured in an acidic condition in the presence of potassium iodide [49]. The concentration of NO_2_^−^ was measured in heart tissue by the Griess reagent method, as previously described [5]. Quantification of free protein thiol groups was determined by derivatization with 5,5′-dithiobis(2-nitrobenzoic acid) (DTNB) by the Ellman method [50]. The concentration of O_2_^−^ in heart tissue was measured at 530 nm after the reaction of nitro blue tetrazolium in TRIS buffer [51]. Determination of PAB in heart tissue was performed using 3,3′, 5,5′-tetramethylbenzidine as a chromogen [52]. Myeloperoxidase enzyme activity heart samples were determined by the o-dianisidine-H_2_O_2_ method, as described by Kothari et al. [13].

The activities of antioxidant enzymes, SOD, CAT, GR and GPx, were measured in heart homogenates by the spectrophotometric method, as previously described [53,54,55,56].

All spectrophotometric analyses of the heart redox state and antioxidant enzyme activities were performed with an Ultrospec 3300 pro UV/Visible spectrophotometer (Amersham Biosciences Corp., Piscataway, NJ, USA).

### 4.4. Histopathological Examination

The heart was removed, and tissue slices were fixed in 10% neutral-buffered formaldehyde, embedded in paraffin and sectioned. The sections, 2 μm thick, were stained by the hematoxylin and eosin (H&E) method. From each specimen, the visual fields, magnified by 20× and 40× were analyzed by using a light microscope (Olympus-2, Tokyo, Japan). Semiquantitative analysis, as we previously described using the modified Billingham scale [57], assessed the degree of the tissue’s damage according to a five-point semiquantitative scale (Table 2). Eight tissue samples from each group were available, and five sections from each tissue samples were analyzed. All pathohistological examinations were performed by two independent observers as a blind study with no prior knowledge of the treatment given to the animals. The sum of these changes represented the Cardiomyocyte Damage Score (CDS) [57].

### 4.5. Immunohistochemistry Examination

Immunostaining was applied on 5 μm thick paraffin sections. After deparaffinization and rehydration, the sections were treated by microwave for 20 min at 400 Win citrate buffer (pH 6.0). After antigen retrieval, samples were incubated overnight with primary antibody for TGF-β (Millipore, MAB1032, TB21; dilution 1:1000), Bax (Millipore, Billerica, MA, USA, dilution 1:250) and Bcl-2 (Millipore, Billerica, MA, USA, dilution 1:200). Sections were then treated with EnVisionTMDetection System (DAKO, GmbH, Jena, Germany) using a biotinylated secondary antibody. Sections from the testicle, serving as a positive control, were processed at the same time, and handled in the same way. Negative controls were performed by omitting the first antibody. The slides were evaluated using a light microscope (Olympus-2, Tokyo, Japan). Slides were evaluated by two of the authors, unaware of immunohistochemical or clinical data using ImageJ, a public domain, Java-based image processing program developed at the National Institute of Health, Bethesda, Maryland, USA.

### 4.6. Statistical Analysis

The data were analyzed by the statistical software package STATISTICA 12, StatSoft Inc, (Tulsa, OK, USA). Results are expressed as mean ± S.D. One way analysis of variance (ANOVA) was applied as appropriate. The differences between examined groups were considered significant if *p* < 0.05.

## 5. Conclusions

In conclusion, resveratrol, apart from its well-characterized activities as an antioxidant, significantly induced reduction in blood pressure through the direct and enzyme-linked ROS neutralization. It also protects heart tissue of hypertensive rats by blunting myeloperoxidase and TGF-beta activity and suppressing the apoptotic process. While conventional therapies targeting control blood pressure are undoubtedly useful, newer approaches, including natural products such as resveratrol, may provide additional benefits against general and malignant hypertension. Resveratrol has successfully been shown to impart protective effects through various mechanisms, thereby making its a potential therapeutic agent for managing severe forms of hypertension.

## Figures and Tables

**Figure 1 ijms-22-05006-f001:**
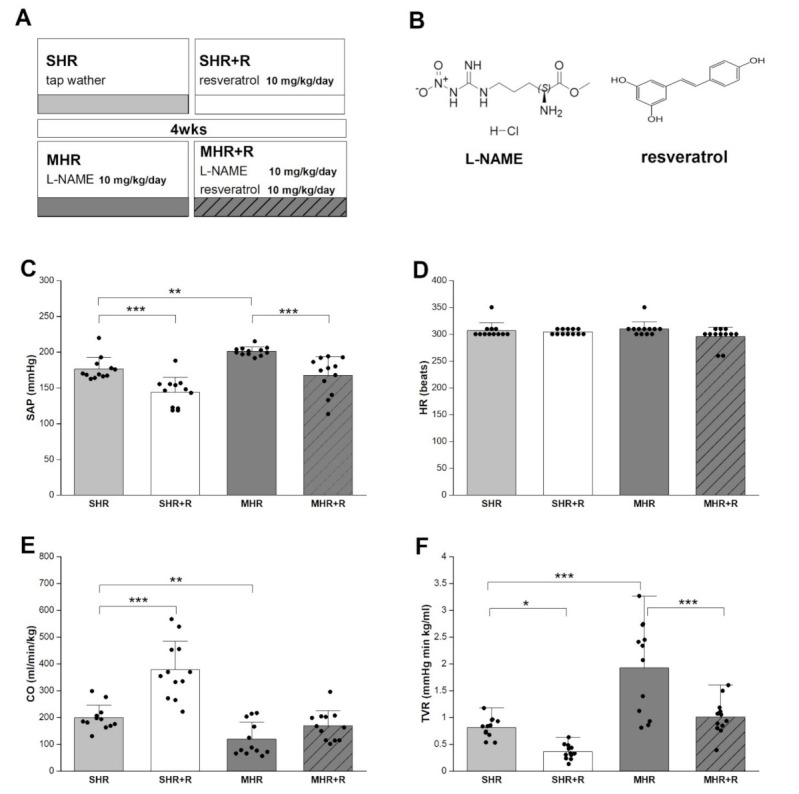
Graphical experimental layout (**A**), chemical formulas (**B**), systolic arterial pressure-SAP (**C**), heart rate-HR (**D**), cardiac output-CO (**E**) and total vascular resistance-TVR (**F**) in experimental groups. SHR-spontaneously hypertensive rat; SHR + R-spontaneously hypertensive rat treated with resveratrol; MHR-malignant hypertensive rat; MHR + R-malignant hypertensive rat treated with resveratrol. Values are means ± SD. ***, **, * indicate *p* < 0.001, 0.01, 0.05.

**Figure 2 ijms-22-05006-f002:**
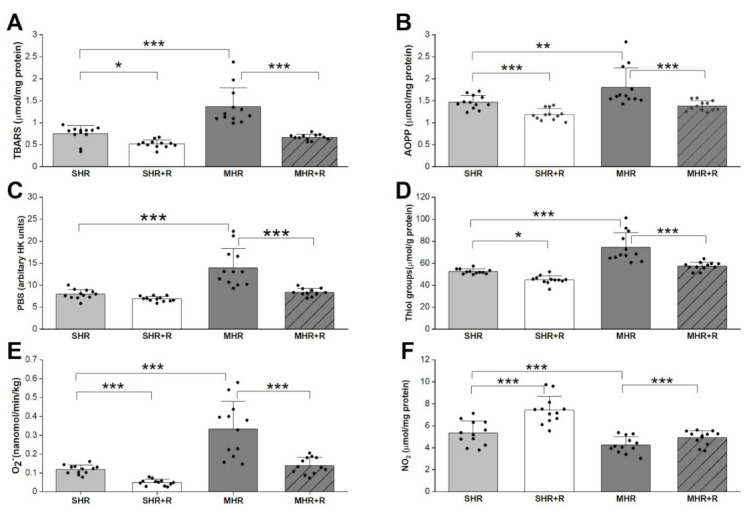
TBARSs-thiobarbituric acid reactive substances (**A**), AOPP-advanced oxidation protein products (**B**), PAB-prooxidative balans (**C**), Thiol groups (**D**), O_2_^−^-superoxide anion radical (**E**), NO_2_^−^-nitrites (F) in experimental groups. SHR-spontaneously hypertensive rat; SHR + R-spontaneously hypertensive treated with resveratrol; MHR-malignant hypertensive rat; MHR + R-malignant hypertensive rat treated with resveratrol. Values are means ± SD. ***, **, * indicate *p* < 0.001, 0.01, 0.05.

**Figure 3 ijms-22-05006-f003:**
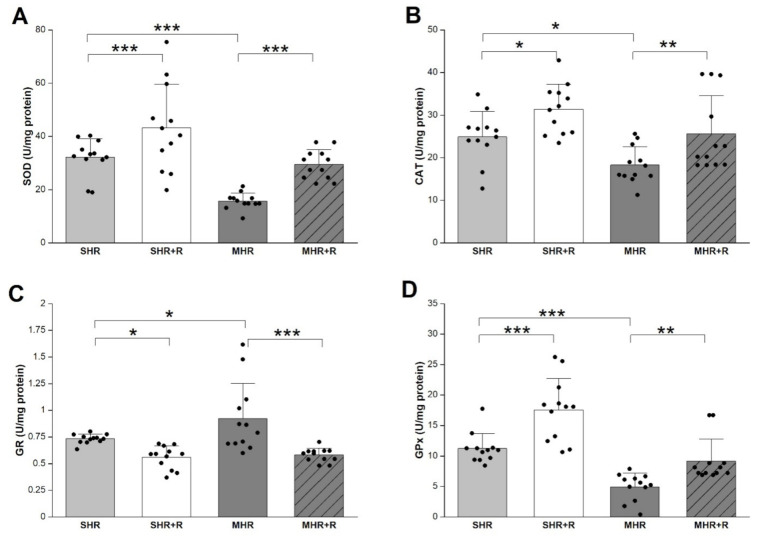
Antioxidative enzyme SOD-superoxide dismutase (**A**); CAT-catalase (**B**); GR-glutathione reductase (**C**) and GPx-glutathione peroxidase (**D**) in experimental groups. SHR-spontaneously hypertensive rat, SHR + R-spontaneously hypertensive treated with resveratrol; MHR-malignant hypertensive rat; MHR + R-malignant hypertensive rat treated with resveratrol. Values are means ± SD. ***, **, * indicate *p* < 0.001, 0.01, 0.05.

**Figure 4 ijms-22-05006-f004:**
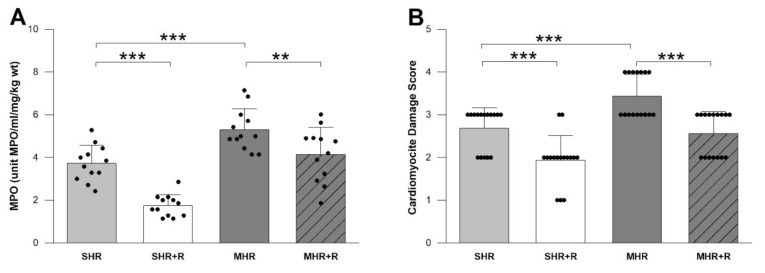
MPO-myeloperoxidase enzyme activity (**A**) and CDS-Cardiomyocyte Damage Score (**B**) in experimental groups. SHR-spontaneously hypertensive rat; SHR + R-spontaneously hypertensive treated with resveratrol; MHR-malignant hypertensive rat; MHR + Rmalignant hypertensive rat treated with resveratrol. Values are means ± SD. *** indicate *p* < 0.001, ** *p* < 0.01.

**Figure 5 ijms-22-05006-f005:**
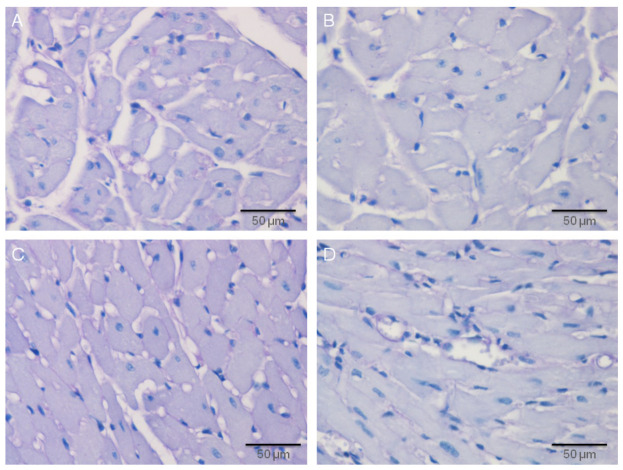
Protective effects of resveratrol treatment in the heart tissue of the essential and malignant hypertensive rats, Hematoxylin and Eosin (H&E) method, 200× magnified images. (**A**) Diffuse vacuolation of the myocytes in the heart tissue of SHR, (**B**), vacuolar degeneration of the myocytes in the heart tissue of MHR, (**C**) mild hypertrophy of the myocytes in the heart tissue of SHR + R, and (**D**) macronodular vacuolation of the myocytes in the heart tissue of MHR + R.

**Figure 6 ijms-22-05006-f006:**
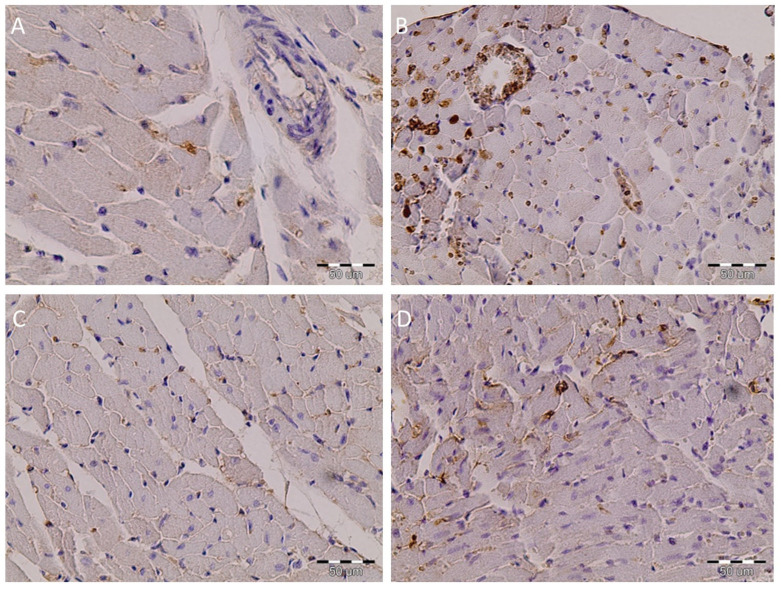
Resveratrol inhibited inflammation in the heart tissue of the essential and malignant hypertensive rats. The expression of TGF-β in heart tissue was examined by immunohistochemical staining, magnification 400×. (**A**) Granular appearance of TGF-β particular in SHR, (**B**) intense cytoplasmic staining in heart tissue in MHR, (**C**) negative immunostaining in the SHR + R group, and (**D**) significant reduction in TGF-β expression in MHR + R group.

**Figure 7 ijms-22-05006-f007:**
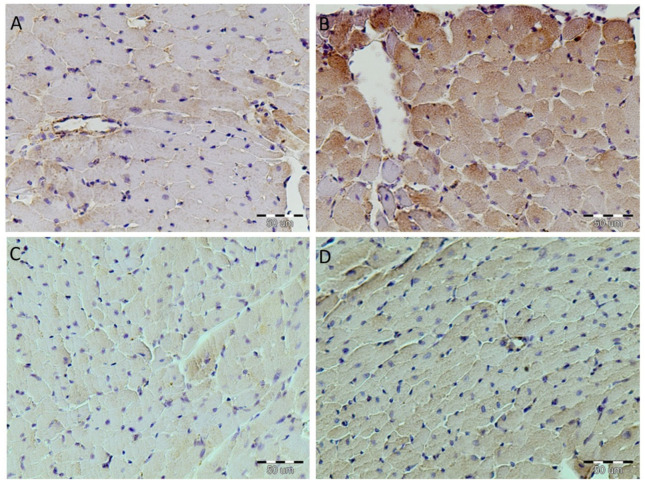
Resveratrol attenuated Bax expression in the heart tissue of the essential and malignant hypertensive rats. The expression of Bax in heart tissue was examined by immunohistochemical staining, magnification 400×. (**A**) The relative number of brown stained myocytes in SHR group, (**B**) expression significantly increased, determined as intensive brown cytoplasmic staining widely distributed in heart tissue of the MHR, (**C**) negative immunostaining in the SHR + R group, and (**D**) myocytes showed rare immuno-positive cells in MHR + R.

**Figure 8 ijms-22-05006-f008:**
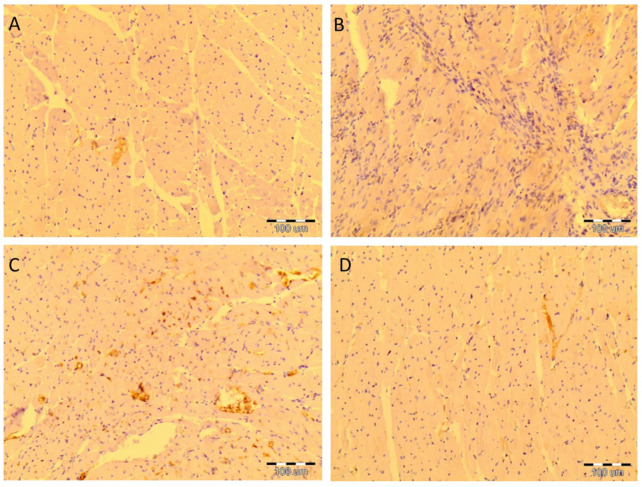
Resveratrol increased Bcl-2 expression in heart tissue of the essential and malignant hypertensive rats. The expression of Bcl-2 in heart tissue was examined by immunohistochemical staining, magnification 400×, positive reaction—yellow color. (**A**) The SHR group showed rare immuno-positive cells, (**B**) negative immunostaining in the MHR, (**C**) intense cytoplasmic staining of Bcl-2 in SHR + R group (yellow), and (**D**) the MHR + R group showed rare immuno-positive cells.

**Table 1 ijms-22-05006-t001:** Left and right ventricular (LV and RV) weight and weight indexes (LVI and RVI).

	SHR	SHR + R	MHR	MHR + R
b.w. (g)	172.5 ± 3.11	147.5 ± 3.92 ***	152.08 ± 1.89 ***	162.92 ± 3.28 *^, #, $^
LV (g)	0.631 ± 0.012	0.526 ± 0.006 ***^, ###^	0.682 ± 0.02 **	0.625 ± 0.008 ^###, $$$^
RV (g)	0.130 ± 0.004	0.108 ± 0.004 ***^, ###^	0.159 ± 0.005 ***	0.122 ± 0.004 ^###, $^
LVI (mg/g)	0.366 ± 0.011	0.337 ± 0.008 ^###^	0.438 ± 0.009 ***	0.413 ± 0.016 **^, $$$^
RVI (mg/g)	0.085 ± 0.006	0.069 ± 0.002 *^, #^	0.085 ± 0.003	0.081 ± 0.004

SHR—spontaneously hypertensive rat, SHR + R—spontaneously hypertensive rat treated with resveratrol; MH—malignant hypertensive rat; MHR + R—malignant hypertensive rat treated with resveratrol. Values are means ± SEM. ***, **, * indicate *p* < 0.001, 0.01, 0.05 vs. SHR; ^###^, ^#^ indicate *p* < 0.001, 0.05 vs. MHR; ^$$$^, ^$^ indicate *p* < 0.001, 0.05 vs. SHR + R group.

**Table 2 ijms-22-05006-t002:** Tissue damage score for the heart alterations (CDS—Cardiomyocyte Damage Score).

Degree	Description
0	Normal histological structure of the heart.
1	A slight loss of transverse striation was observed in individual cells.
2	Smaller groups of cells with a pronounced loss of transverse striation and the appearance of small vacuoles in the sarcoplasm.
3	A larger number of cells with a pronounced loss of transverse striation and the appearance of large vacuoles in the sarcoplasm.
4	Diffuse vacuolar degeneration of cardiomyocyte cytoplasm followed by vacuolar degeneration.

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
