# Peer review of "Resveratrol Protects Cardiac Tissue in Experimental Malignant Hypertension Due to Antioxidant, Anti-Inflammatory, and Anti-Apoptotic Properties"

_ijms, 2021, doi:10.3390/ijms22095006_

Round 1

Reviewer 1 Report

I read with interest the article about the positive effects of resveratrol on the cardiac tissue. In my opinion, the article is well planned, properly described and prepared. I believe that the article should be published after minor revision.

Comments:

  • Abstract should follow the style of structured abstracts, but without headings: 1) Background; 2) Methods; 3) Results; and 4) Conclusion
  • Authors should improve the quality of axis descriptions in the figures.
  • Figure 3 – B, D – The unit of enzyme activity should be capitalized
  • Page 13, line 259 – should be Eiserich el al.
  • Page 13, line 279 – should be Hockenbery et al.
  • There are double spaces in the text at certain points. Dots are missing at the end of a few sentences.
  • The name of the program that was used for the statistics should be added.
  • Bibliography should be corrected (e.g. no. 4, 5, 8, etc – lack of Volume and page range.)

Author Response

Dear Reviewer,

before all, we want to thank you for all the comments and very constructive suggestions that have contributed to the better quality of our manuscript. We accepted the reviewers΄ suggestions and made changes in the main text in accordance with the instructions for authors, with the given explanation.

We are looking forward to hearing your final decision.

Yours sincerely,

Prof. Dr Jelica Grujic-Milanovic, PhD.

on the behalf of all co-authors

Reviewer 2 Report

Hypertension, one of the most common medical disorders, is associated with an increased incidence of all-cause and cardiovascular disease (CVD) mortality. Although hypertension affects a large proportion of the population, its etiology remains poorly defined. Resveratrol is a polyphenol that is abundant in grape skin and seeds. Previous studies have shown that resveratrol can play a beneficial role in the prevention and in the progression of chronic diseases related to inflammation, regulates immunity by interfering with immune cell regulation.

In this study, Grujic-Milanovic et al aimed to examine the effects of resveratrol on blood pressure, oxidative stress, and inflammation, as well as cardiac structure in two models of experimental hypertension, spontaneously (SHR) and malignant hypertensive (MHR) rats. They showed that resveratrol administration decreased blood pressure in both SHR and MHR rats. Resveratrol treatment ameliorated morphological changes of the heart tissue, improved both hemodynamics and histopathology parameters. The authors further concluded that the beneficial effects of resveratrol could be due to its antioxidant, anti-inflammatory, and anti-apoptotic properties.

This study is very interesting and has high pathophysiological relevance as there is an urgent need for the treatment of hypertension. This study will thus offer new insights on improving the treatment of hypertension. The findings in this manuscript were innovative and will have an impact on the field.

However, besides the IHC images of TGF-β, Bax, and Bcl-2, that would be a plus if you could quantify these IHCs. Alternatively, western blots on these proteins would be convincing too.

Author Response

Dear Reviewer,

Before all, we want to thank you for all the comments and very constructive suggestions that have contributed to the better quality of our manuscript. We accepted the reviewers΄ suggestions and made changes in the main text in accordance with the instructions for authors, with the given explanation.

We are looking forward to hearing your final decision.

Yours sincerely,

Prof. Dr Jelica Grujic-Milanovic, PhD.

on the behalf of all co-authors

Reviewer 3 Report

In the manuscript by Jelica Grujic-Milanovic et al., the authors used hypertensive rats to investigate the effects of resveratrol on oxidative stress and inflammation. Also, they examined the resveratrol effects on cardiac structure and function. The topic is of interest, nevertheless, there are several limitations in the manuscript. Comments are as follows:

- There are some typos and editing errors in the manuscript that detract in a significant way from the readability of the manuscript. The manuscript should be carefully edited before considered for publication. For example, (1) Line 264: the word “trough”. (2) Abbreviations (e.g., TVR, TBARS, SHR) and their full form should be provided wherever they appear first time in the manuscript and later use abbreviation.

- Animal experimental layout is desired for a better understanding of the animal study.

- Authors used NOS antagonist to induce MHR. This is not mentioned in the method section and thus the purpose of using L-NAME should be brief discussed in the method section.

- Authors used a decent number of animals (n=12/group) in the study. Mean ± SEM looks “tighter” in the groups in almost all the figures. I would like to see all the figures in the bar graphs overlaid with a dot plot and use mean ± SD.

- In the study, authors compared MHR with SHR-R. I do not find this rationale providing multiple comparisons in statistical analysis. Also, statistical comparison between the groups can be shown using a line.

 - What is the rationale for using female SHR rats while the prevalence of hypertension is reflected in both males and females?

- NADPH oxidases are an important source of ROS and responsible for the formation of superoxide anion, it would be interesting to see the effect of resveratrol on NADPH oxidase activity.

- It is desirable to perform the experiments investigating molecular mechanisms underlying the antioxidative, anti-inflammatory, and anti-apoptotic effect of resveratrol in a hypertensive rat model.

- Hypertension is often associated with stroke and renal failure. Did the authors investigate any structural and functional abnormalities in the kidney and brain following resveratrol treatment?

Author Response

In the manuscript by Jelica Grujic-Milanovic et al., the authors used hypertensive rats to investigate the effects of resveratrol on oxidative stress and inflammation. Also, they examined the resveratrol effects on cardiac structure and function. The topic is of interest, nevertheless, there are several limitations in the manuscript. Comments are as follows:

Dear Reviewers,

We want to thank you for all the comments and very constructive suggestions that would contribute to the better quality of our manuscript.

Please accept our apologies for the confusion caused by delay in communication.

Namely, we received the information from the IJMS on April 1st that there were only two reviewer’s reports, and we replied on them on April 4th. The information about your review was forwarded to us on April 7th, and that is the only reason why we are sending a response to your review now.

There are some typos and editing errors in the manuscript that detract in a significant way from the readability of the manuscript. The manuscript should be carefully edited before considered for publication. For example, (1) Line 264: the word “trough”. (2) Abbreviations (e.g., TVR, TBARS, SHR) and their full form should be provided wherever they appear first time in the manuscript and later use abbreviation.

We corrected the text body in accordance with your suggestions.

Abbreviations (e.g., SHR, L-NAME, CO, TVR, TBARS) and their full form are listed wherever they appear first time in the manuscript and later we use abbreviation.

Please see the lines: 46, 71, 74, 79, 95, 99, 101, 102, 105, 120, 121, 125, 127-128, 144-145, 276, 337.

Animal experimental layout is desired for a better understanding of the animal study.

We replaced Figure 1 with a new one. Also, we included graphical experimental layout in the text body.

Authors used NOS antagonist to induce MHR. This is not mentioned in the method section and thus the purpose of using L-NAME should be brief discussed in the method section.

We corrected the text body in accordance with your suggestions.

Please see the lines: 45, 324-327.

Authors used a decent number of animals (n=12/group) in the study. Mean ± SEM looks “tighter” in the groups in almost all the figures. I would like to see all the figures in the bar graphs overlaid with a dot plot and use mean ± SD.

The authors completely changed Figure 1, Figure 2, Figure 3 and Figure 4 according to your instructions.

In the study, authors compared MHR with SHR-R. I do not find this rationale providing multiple comparisons in statistical analysis. Also, statistical comparison between the groups can be shown using a line.

The authors thank you for your suggestions and done statistical analysis, excluding this comparation.

What is the rationale for using female SHR rats while the prevalence of hypertension is reflected in both males and females?

Rationale for using female SHR lays in a fact that cardiovascular diseases are the leading cause of death in women and that their mortality rate is even higher in comparison to men. The risk of heart disease in women is often underestimated due to the misperception that females are ‘protected’ against cardiovascular disease (1,2).  Also, it was shown that resveratrol reduced blood pressure in males in a several models of experimental hypertension. However, there are limited studies which analyze protective effects of resveratrol on blood pressure in females. Treatment of female SHR with resveratrol may lead to sex-specific preventive strategies and therapeutics use of this polyphenol.

  1. MaasA.H.E.M.; Appelman Y.E.A. Gender differences in coronary heart disease. Neth Heart J. 2010 Dec; 18(12): 598–602. doi: 10.1007/s12471-010-0841-y
  2. Gao Z., Chen Z., Sun A., Deng X. Gender differences in cardiovascular disease. Medicine in Novel Technology and Devices. Volume 4, December 2019, 100025

NADPH oxidases are an important source of ROS and responsible for the formation of superoxide anion, it would be interesting to see the effect of resveratrol on NADPH oxidase activity.

Thank you very much for your interesting notice. We agree that NADPH oxidase has a crucial role in ROS formation. Many authors investigated influence of resveratrol on its activity and showed the ability of NADPH oxidase to affect blood pressure in fructose induced oxidative stress (3,4). Unfortunately, we have no possibility to do new experiments including this interesting topic, especially in a limited response time of 10 days.

  1. Cheng PW., Ho WY.,Su YT., Lu PJ., Chen BZ., Cheng WH., Lu WH, Sun GC., Yeh TC., Hsiao M., Tseng CJ. Resveratrol decreases fructose-induced oxidative stress, mediated by NADPH oxidase via an AMPK-dependent mechanism. Br J Pharmacol. 2014 Jun;171(11): 2739-50.doi: 10.1111/bph.12648.
  2. Yeh TC., Shin CS., Chen HH., Lai CC.,Sun GC, Tseng CJ., Cheng PW. Resveratrol regulates blood pressure by enhancing AMPK signaling to downregulate a Rac1-derived NADPH oxidase in the central nervous system. J Appl Physiol 125: 40–48, 2018. First published March 1, 2018; doi:10.1152/japplphysiol.00686.2017

It is desirable to perform the experiments investigating molecular mechanisms underlying the antioxidative, anti-inflammatory, and anti-apoptotic effect of resveratrol in a hypertensive rat model.

That would be nice but considering that cardiac remodeling and hypertrophy are multifactorial in nature, and that many authors have described molecular mechanisms of resveratrol in cardiopathology so far (4), we attempted to focus our study on association between oxidative stress, inflammation, and apoptosis in the heart of essentially and malignantly hypertensive rats and the ability of resveratrol to prevent such remodeling. Nevertheless, performance of the new study in such short response time (within 10 days) is impossible.

  1. Ramachandra CJA, Cong S, Chan X, Yap EP, Yu F, Hausenloy DJ. Oxidative stress in cardiac hypertrophy: From molecular mechanisms to novel therapeutic targets. Free Radic Biol Med. 2021 Apr; 166:297-312. doi: 10.1016/j.freeradbiomed.2021.02.040. Epub 2021 Mar 4

Hypertension is often associated with stroke and renal failure. Did the authors investigate any structural and functional abnormalities in the kidney and brain following resveratrol treatment?

We agree with your remarks, but we did not investigate such effects of resveratrol now. Considering that 4-week L-NAME administration is associated with decreased glomerular density, increased glomerular tuft area, tubular injury, and profound renal fibrosis in Wistar rats (6), as well as the existence of predisposition of spontaneously hypertensive rats to develop renal injury during nitric oxide synthase inhibition (7), we have already planned to continue our future research into the effects of resveratrol in this model of progressive renal failure.

  1. Stanko P., Baka T., Repova K., Aziriova S., Krajcirovicova K., Barta A., Janega P., Adamcova M., Paulis L., Simko F. Ivabradine Ameliorates Kidney Fibrosis in L-NAME-Induced Hypertension. Front Med (Lausanne). 2020 Jul 10;7: 325.doi: 10.3389/fmed.2020.00325.eCollection 2020.

7. Verhagen M., Koomans H., Joles J. Predisposition of spontaneously hypertensive rats to develop renal injury during nitric oxide synthase inhibition. European Journal of Pharmacology 411, 2001. 175–180

Round 2

Reviewer 3 Report

The authors' response is missing. The revision request is not taken care of in the revised version of the manuscript.

Author Response

(The authors gave the same response as above.)

Round 3

Reviewer 3 Report

I could only see the comments from the authors but could not find the updated version of the manuscript where the authors claimed to perform the changes as I suggested. There are the following version of the manuscript is available for review: v1 is the first version, and v2 is not significantly changed. Authors need to upload the most updated version of the manuscript in which they executed the changes for review.

Author Response

In the manuscript by Jelica Grujic-Milanovic et al., the authors used hypertensive rats to investigate the effects of resveratrol on oxidative stress and inflammation. Also, they examined the resveratrol effects on cardiac structure and function. The topic is of interest, nevertheless, there are several limitations in the manuscript. Comments are as follows:

Dear Reviewers,

We want to thank you for all the comments and very constructive suggestions that would contribute to the better quality of our manuscript.

Please accept our apologies for the confusion caused by delay in communication.

Namely, we received the information from the IJMS on April 1st that there were only two reviewer’s reports, and we replied on them on April 4th. The information about your review was forwarded to us on April 7th, and that is the only reason why we are sending a response to your review now.

- There are some typos and editing errors in the manuscript that detract in a significant way from the readability of the manuscript. The manuscript should be carefully edited before considered for publication. For example, (1) Line 264: the word “trough”. (2) Abbreviations (e.g., TVR, TBARS, SHR) and their full form should be provided wherever they appear first time in the manuscript and later use abbreviation.

We corrected the text body in accordance with your suggestions.

Abbreviations (e.g., SHR, L-NAME, CO, TVR, TBARS) and their full form are listed wherever they appear first time in the manuscript and later we use abbreviation.

Please see the lines: 46, 71, 74, 79, 95, 99, 101, 102, 105, 120, 121, 125, 127-128, 144-145, 276, 337.

- Animal experimental layout is desired for a better understanding of the animal study.

We replaced Figure 1 with a new one. Also, we included graphical experimental layout in the text body.

- Authors used NOS antagonist to induce MHR. This is not mentioned in the method section and thus the purpose of using L-NAME should be brief discussed in the method section.

We corrected the text body in accordance with your suggestions.

Please see the lines: 45, 324-327.

- Authors used a decent number of animals (n=12/group) in the study. Mean ± SEM looks “tighter” in the groups in almost all the figures. I would like to see all the figures in the bar graphs overlaid with a dot plot and use mean ± SD.

The authors completely changed Figure 1, Figure 2, Figure 3 and Figure 4 according to your instructions.

- In the study, authors compared MHR with SHR-R. I do not find this rationale providing multiple comparisons in statistical analysis. Also, statistical comparison between the groups can be shown using a line.

The authors thank you for your suggestions and done statistical analysis, excluding this comparation.

 - What is the rationale for using female SHR rats while the prevalence of hypertension is reflected in both males and females?

Rationale for using female SHR lays in a fact that cardiovascular diseases are the leading cause of death in women and that their mortality rate is even higher in comparison to men. The risk of heart disease in women is often underestimated due to the misperception that females are ‘protected’ against cardiovascular disease (1,2).  Also, it was shown that resveratrol reduced blood pressure in males in a several models of experimental hypertension. However, there are limited studies which analyze protective effects of resveratrol on blood pressure in females. Treatment of female SHR with resveratrol may lead to sex-specific preventive strategies and therapeutics use of this polyphenol.

  1. MaasA.H.E.M.; Appelman Y.E.A. Gender differences in coronary heart disease. Neth Heart J. 2010 Dec; 18(12): 598–602. doi: 10.1007/s12471-010-0841-y
  2. Gao Z., Chen Z., Sun A., Deng X. Gender differences in cardiovascular disease. Medicine in Novel Technology and Devices. Volume 4, December 2019, 100025

- NADPH oxidases are an important source of ROS and responsible for the formation of superoxide anion, it would be interesting to see the effect of resveratrol on NADPH oxidase activity.

Thank you very much for your interesting notice. We agree that NADPH oxidase has a crucial role in ROS formation. Many authors investigated influence of resveratrol on its activity and showed the ability of NADPH oxidase to affect blood pressure in fructose induced oxidative stress (3,4). Unfortunately, we have no possibility to do new experiments including this interesting topic, especially in a limited response time of 10 days.

  1. Cheng PW., Ho WY.,Su YT., Lu PJ., Chen BZ., Cheng WH., Lu WH, Sun GC., Yeh TC., Hsiao M., Tseng CJ. Resveratrol decreases fructose-induced oxidative stress, mediated by NADPH oxidase via an AMPK-dependent mechanism. Br J Pharmacol. 2014 Jun;171(11): 2739-50.doi: 10.1111/bph.12648.
  2. Yeh TC., Shin CS., Chen HH., Lai CC.,Sun GC, Tseng CJ., Cheng PW. Resveratrol regulates blood pressure by enhancing AMPK signaling to downregulate a Rac1-derived NADPH oxidase in the central nervous system. J Appl Physiol 125: 40–48, 2018. First published March 1, 2018; doi:10.1152/japplphysiol.00686.2017

- It is desirable to perform the experiments investigating molecular mechanisms underlying the antioxidative, anti-inflammatory, and anti-apoptotic effect of resveratrol in a hypertensive rat model.

That would be nice but considering that cardiac remodeling and hypertrophy are multifactorial in nature, and that many authors have described molecular mechanisms of resveratrol in cardiopathology so far (4), we attempted to focus our study on association between oxidative stress, inflammation, and apoptosis in the heart of essentially and malignantly hypertensive rats and the ability of resveratrol to prevent such remodeling. Nevertheless, performance of the new study in such short response time (within 10 days) is impossible.

  1. Ramachandra CJA, Cong S, Chan X, Yap EP, Yu F, Hausenloy DJ. Oxidative stress in cardiac hypertrophy: From molecular mechanisms to novel therapeutic targets. Free Radic Biol Med. 2021 Apr; 166:297-312. doi: 10.1016/j.freeradbiomed.2021.02.040. Epub 2021 Mar 4

- Hypertension is often associated with stroke and renal failure. Did the authors investigate any structural and functional abnormalities in the kidney and brain following resveratrol treatment?

We agree with your remarks, but we did not investigate such effects of resveratrol now. Considering that 4-week L-NAME administration is associated with decreased glomerular density, increased glomerular tuft area, tubular injury, and profound renal fibrosis in Wistar rats (6), as well as the existence of predisposition of spontaneously hypertensive rats to develop renal injury during nitric oxide synthase inhibition (7), we have already planned to continue our future research into the effects of resveratrol in this model of progressive renal failure.

  1. Stanko P., Baka T., Repova K., Aziriova S., Krajcirovicova K., Barta A., Janega P., Adamcova M., Paulis L., Simko F. Ivabradine Ameliorates Kidney Fibrosis in L-NAME-Induced Hypertension. Front Med (Lausanne). 2020 Jul 10;7: 325.doi: 10.3389/fmed.2020.00325.eCollection 2020.
  2. Verhagen M., Koomans H., Joles J. Predisposition of spontaneously hypertensive rats to develop renal injury during nitric oxide synthase inhibition. European Journal of Pharmacology 411, 2001. 175–180